# A Termination Criterion for Probabilistic Point Clouds Registration

**Simone Fontana** *[iD] and **Domenico Giorgio Sorrenti** [iD]

Department of Informatics, Systems and Communication, Università degli Studi di Milano-Bicocca, 20126 Milano, Italy; domenico.sorrenti@unimib.it
* Correspondence: simone.fontana@unimib.it

**Abstract:** Probabilistic Point Clouds Registration (PPCR) is an algorithm that, in its multi-iteration version, outperformed state-of-the-art algorithms for local point clouds registration. However, its performances have been tested using a fixed high number of iterations. To be of practical usefulness, we think that the algorithm should decide by itself when to stop, on one hand to avoid an excessive number of iterations and waste computational time, on the other to avoid getting a sub-optimal registration. With this work, we compare different termination criteria on several datasets, and prove that the chosen one produces very good results that are comparable to those obtained using a very large number of iterations, while saving computational time.

**Keywords:** point cloud registration; point set; ICP; alignment

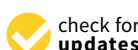



## 1. Introduction

Point clouds registration is the problem of finding the transformation (mostly a rigid transformation) that best aligns two point clouds, usually called the source and target point clouds.

One of the first approaches to this problem, and still one of the most used, is Iterative Closest Point (ICP) [1–3], which aligns two point clouds by minimizing the sum of distances between corresponding points, where corresponding points are nearest neighbouring points. Probabilistic Point Clouds Registration (PPCR) [4] is a variant of ICP that uses a probabilistic model to improve the robustness against noise and outliers, one of the most relevant problem of local registration algorithms. Much like ICP, it is an iterative algorithm that repeatedly tries to improve a solution, until a stopping criterion is satisfied.

The experiments show that it outperformed most state-of-the-art local registration algorithms in this field. However, these experiments have been performed using a large fixed number of iterations as stopping criterion. Instead, we think that, to be of practical utility, an iterative algorithm should autonomously decide when to stop. Indeed, using a fixed number of iterations on one hand does not guarantee that the best solution have been found; on the other, it could result in an excess of computation time, because the solution have been found earlier.

Despite the relevance of the topic, very little research has been conducted on finding the best termination criterion for point clouds registration algorithms. Moreover, many commonly used techniques have parameters to fine tune to each specific application [5].

For these reasons, we propose an improvement of PPCR, analyzing different termination criteria and finding the best one. Furthermore, we demonstrate that the chosen solution is as effective as using a very large number of iterations but, at the same time, results in fewer iterations and, therefore, less computational time. The experiments have been conducted on the IRALab Point Clouds Registration Benchmark [6], which is composed of many different point clouds from different environments. We used the same set of parameters for every experiment, to prove that it is generic enough that it does not need to be fined tuned to different environments.

## 2. Related Work

Point clouds registration algorithms could be divided into two categories: global and local.

Global registration is the problem of aligning two point clouds without any prior assumption on their misplacement. Traditionally, this problem has been solved using feature-based techniques, such as PFH [7] and their faster variant FPFH [8], or angular-invariant features, [9]. Usually the matches are found using algorithms such as RANSAC [10]; the matches are then used to estimate the rototranslation between the two point clouds. As an alternative to *hand-crafted* descriptors, solutions based on neural networks, that aim at enhancing the discriminative capacity of the features, have been proposed. Examples are 3dMatch [11] and 3DSmoothNet [12]. Networks that combines both the feature matching and the transformation estimation steps together have been proposed too, such as Pointnetlk [13] and Pcrnet [14].

The drawback of global registration approaches is that they usually cannot provide an accurate alignment, mainly because of the high number of spurious matches; therefore, they are rather used to obtain a coarse registration that is later refined with a fine registration algorithm [5]. For this reason, techniques aimed at estimating a rototranslation from matches with a high number of outliers have been proposed. Notable examples are Fast Global Registration [15] and TEASER++ [16], that can even work without any feature, but using an all-to-all association strategy.

Local registration algorithms, instead, (also known as *fine* registration) aim at finding the rototranslation that best aligns two point cloud that are already roughly aligned. Therefore, they refine a pre-existing alignment, that can be obtained in different ways, for example, with a global algorithm, with an inertial system, or manually.

One of the most important algorithms in this category is Iterative Closest Point (ICP). ICP was developed independently by Besl and McKay [1], Chen and Medioni [2], and Zhang [3] and is still one of the most used technique. The most critical problem a registration algorithm has to solve is the data association problem, that is, associating one point in a point clouds, to one or more in the another. ICP solves this issue by associating a point in the source point cloud to the closest in the target. The best transformation resulting from this data association is found and this process is repeated until convergence.

Many different variants of ICP have been proposed. Usually, they aim at speeding up the algorithm or at improving the accuracy [17]. One of the most important of these variants is Generalized ICP (G-ICP) [18], which greatly improves the quality of the results by using a probabilistic framework with a point-to-plane data association.

Probabilistic Point Clouds Registration (PPCR) [4] uses the same closest-point based data association of ICP, in conjunction with a probabilistic model, to improve both the accuracy and, most important, the robustness against noise and outliers. While it was originally developed to deal with the problem of aligning a sparse point cloud with a dense one, it was shown to perform very well also on traditional registration problems.

Another important technique used for local point clouds registration is called Normal Distribution Transform (NDT) [19]. This technique was originally developed to register 2D laser scans, but has been successfully applied also to 3D point clouds [20]. Differently from ICP, it does not establish any explicit correspondence between points. Instead, the source point cloud or laser scan is subdivided into cells and a normal distribution is assigned to each cell, so that the points are represented by a probability distribution. The matching problem is then solved as a maximization problem, using Newton's algorithm.

ICP-like algorithms are usually iterative, that is, they perform several iterations, each composed of an optimization problem that should improve the previous solution. Deciding when to stop the algorithm is an important task, since stopping too early would result in a sub-optimal solution, while too late would be a waste of computational time (crucial in real-time robotics applications). Moreover, since these algorithms use heuristics to estimate the data association, it may happen that using too many iterations worsen the result, as we show in the experimental section.

Common termination criteria used for point clouds registration algorithms includes: a maximum number of iterations, a relative or absolute transformation threshold or Mean Squared Error (MSE) threshold, and a maximum number of similar iterations [5]. Despite the relevance, very little research has been conducted on this topic. To our knowledge, no systematic comparison of different termination criteria exists. Moreover, commonly used criteria have parameters that need to be carefully tuned for each specific scenario.

However, point clouds registration has several affinities to approaches to optimization based on heuristics, such as Particle Swarm Optimization (PSO) [21] or Genetic Algorithms. Indeed, it is typically framed as an optimization over the relative pose between the clouds, exploiting some kind of data association heuristic. For this reason, we took inspiration from the literature on termination criteria for other approaches to optimization. For example, Padmanabhan et al. [22] proposed a novel termination criterion for least square optimization which does not depend on a prior knowledge of the optimum and which does not heavily relies on fine tuned parameters. Even though the application is different, it shares some important goals with our work. Zielinski et al. [23] derived a termination criterion for evolutionary algorithms that analyzes the population to automatically decide when to stop the optimization process; although it is a sensible alternative to using a fixed number of iterations, it still relies on a set of parameters that depends on the specific optimization problem. Another common approach used with evolutionary algorithms is to obtain an upper bounds on the number of iterations required for an optimal solution, with a certain probability [24,25].

## 3. Materials and Methods

### 3.1. Probabilistic Point Clouds Registration

We already presented PPCR in a previous work [4]; however, since we present an extension to the original version, we briefly summarize its working.

PPCR is a closest-point based algorithm for local point clouds registration. This means that it is aimed at fine-aligning two point clouds that are already roughly aligned. It does not use any feature to estimate correspondences between two point clouds; instead, similarly to ICP, it approximates the true, unknown, correspondences by using a data-association policy based on the closest distance.

However, the PPCR data association policy differs from that of ICP (and many of its variants)—in ICP each point in the source point cloud is associated with only a single point in the target point cloud, while PPCR associates a point in the source point cloud with a set of points in the target cloud. Moreover, each association is weighted. The weights represent the probability of an association of being the right data-association for a particular point.

The two different data association methods are depicted in Figure 1.

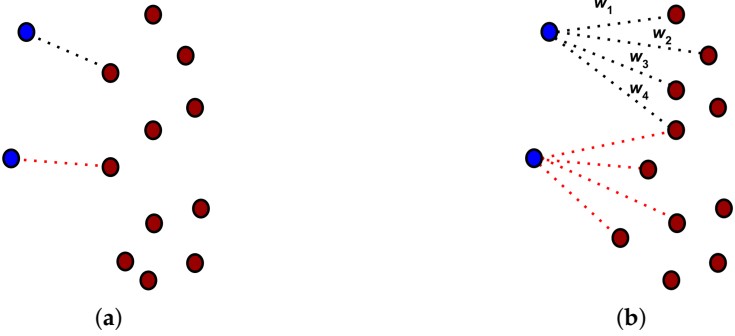

(**a**)             (**b**)

**Figure 1.** The two different data association policies. (**a**) Iterative Closest Point (ICP) Data Association (**b**) Probabilistic Data Association

For each point $x_j$ in the source point cloud, we look for the $n$ nearest points, $y_0, ..., y_n$, in the target cloud. For each of these points $y_k$, with $0 \leq k \leq n$, we define an error term given by

$$\|y_k - (Rx_j + T)\|^2, \tag{1}$$

where $R$ is a rotation matrix and $T$ a translation vector. Equation (1) represents the squared error between the point $y_k$ in the target point cloud and the associated point $x_j$ from the source point cloud, transformed using the current estimate of the rotation $R$ and translation $T$.

Summing all the error terms, calculated according to Equation (1), we build a Least Squares optimization problem which is solved using a suitable method (such as Levenberg-Marquardt). However, given a set of points associated to $x_j$, not all the corresponding error terms should have the same weight. Intuitively we want to give more importance to the associations which are in accordance with the current estimate of the transformation and lower importance to the others. Thus, the weight $w_{kj}$ of the error term $\|y_k - (Rx_j + T)\|^2$ is given by

$$w_{kj} \propto e^{-\frac{\|y_k - (Rx_j+T)\|^2}{2}}, \tag{2}$$

where the proportionality implies a normalization among all the error terms associated with $x_j$ so that their weights represent a probability distribution (therefore they must sum to 1). Equation (2) was derived from an Expectation-Maximization algorithm [26], with an additive Gaussian noise model [4].

The Gaussian probability density function (pdf) used in Equation (2) is appropriate assuming that there are no outliers and all points in the source point cloud have a corresponding point in the target point cloud. However, a t-distribution is a better choice in presence of outliers, especially when there is lot of distortion in one of the point clouds that, thus, cannot be aligned perfectly. Consequently, we decided to use a more robust formulation for the weights, basing on the t-distribution. A t-distribution is very similar to a Gaussian, but its tails have a higher probability; therefore, it better represents a population with a higher probability of having outliers.

The weight $w_{kj}$ of the association between $x_j$ and $y_k$ is given by:

$$w_{kj} = p_{kj}\frac{\nu + d}{\nu + \|y_k - (Rx_j + T)\|^2}, \tag{3}$$

with

$$p_{kj} \propto \left(1 + \frac{\|y_k - (Rx_j + T)\|^2}{\nu}\right)^{-\frac{\nu+d}{2}}, \tag{4}$$

where $\nu$ the is number of degrees of freedom of the t-distribution (which is a parameter of our algorithm) and $d$ is the dimension of the error terms (in our case 3, since we are operating with points in the 3D space). Equations (3) and (4) were derived from the same Expectation-Maximization model of Equation (2), but using a multivariate t-distribution instead of a Gaussian pdf. For the full demonstration of the derivation, please consult the original paper [4].

In Equations (1)–(4) we need an estimate of the rotation and translation; however, these are estimated by solving the optimization problem whose error terms are weighted with the weights we want to calculate. Hence, our problem cannot be formulated as a simple least-square error problem, but it has to be reformulated as an Expectation Maximization problem. During the Expectation phase, the latent variables, that is, the weights, are estimated using the previous iteration estimate of the target variables (the rotation and translation); during the Maximization phase, the problem becomes a least-square error optimization problem, with the latent variables assuming the values estimated during the Expectation phase.

The proposed approach, in its multi-iteration version, is composed of two nested loops. The inner one finds the best rototranslation that minimizes the sum of weighted

squared errors (as in Equation ([1](#)), very similarly to ICP. However, differently to ICP, our problem cannot be solved in closed form and, thus, we use an iterative algorithm such as Levenberg-Marquard. It has to be noted that, at each iteration of Levenberg-Marquard, the associations are not estimated again, but their weights are recalculated. Thus, we solve an iteratively reweighted mean squared error problem. In the outer loop, we move the source cloud with the result of the optimization, re-estimate the associations and build a new optimization problem.

This structure has been already briefly described in our previous work. However, here we present a novel way to decide when the outer loop should stop, instead than using a predefined number of iterations.

### 3.2. Termination Criteria

When the source and the target point clouds are very close, a single iteration of the proposed probabilistic point clouds registration algorithm may be enough. However, a typical real scenario requires more than a single iteration.

To obtain a good solution, most of the correspondences used to form the optimization problem need to be right. Since we use an ICP-like data association policy based on nearest neighbours, this happens only if the two point clouds are close enough.

In our algorithm, two parameters control which and how many points in the target point cloud are associated to a particular point in the source point cloud—the maximum distance between neighbours and the maximum number of neighbours. Setting these parameters to very high values could help the algorithm to converge to a good solution even when the starting poses of the two point clouds are not really close. However, this solution allows more outliers, that is, wrong data associations, to get into the optimization step. Even tough the probabilistic approach has the capability to soft-filtering out outliers, thanks to the probabilistic weighting technique, using too many points will lead to a huge optimization problem which would be very slow to solve. Usually, a much more practical and fast solution is to use lower values for the maximum distance and the maximum number of neighbors, while using multiple iterations of the probabilistic approach, which implies re-estimating the data associations, in the same way it is done, for example, in ICP and G-ICP.

Using this technique, our approach becomes composed of two nested loops. The outer one moves the source point cloud with the current result, estimates the point correspondences, sets up the optimization problem, and activates the solver. This process is repeated until a convergence criterion is met. The inner loop, instead, is composed of the iterations of the Levenberg-Marquard algorithm, which is used to solve the optimization problem.

The multi iteration version of our algorithm provides good results, compared to other state-of-the-art algorithms [4]. Of course, in order to be of practical usefulness, such an algorithm would greatly benefit from some kind of automatic termination criterion. It would mean that the algorithm could decide by itself when it should stop, without adaptation of parameters to the specifics of each problem.

The most simple termination criterion is to use a fixed predefined number of iterations. This is the technique we used in our previous work. However, this solution is far from being optimal, since the number of iterations would become a parameter of the algorithm. Most importantly, there would be no automatic way of estimating this parameter *a-priori*, so this solution is unpractical and has to be discarded. Lastly, using a fixed value for this parameter would probably mean using too many iterations in some cases and using too few in others. On the other hand, using a very large value would greatly increase the execution time, in many cases without improving the quality of the result.

For these reasons, we evaluated different automatic termination criteria, to find which one works best with PPCR.

Our first choice was to evaluate the Mean Squared Error (MSE) with respect to the previous iteration: we take the source point cloud and apply, separately, the rototranslations estimated during the current iteration of the algorithm and during the previous

one. Therefore, we have the same point cloud in two different poses. Since applying a rototranslation, that is, a matrix multiplication, maintains the order of the points, we know that point $x_i^t$ in $X^t$ (the source point cloud aligned with the current estimate) corresponds to point $x_i^{t-1}$ in $X^{t-1}$ (the source point cloud aligned with the previous estimate). Hence, the point correspondences are known and exact. We used Equation (5), where $N$ is the size of the point cloud, to calculate the Mean Squared Error (MSE) between two iterations.

$$MSE(X^t, X^{t-1}) = \frac{\sum_i^N ||x_i^t - x_i^{t-1}||^2}{N}. \tag{5}$$

We stop the algorithm when the MSE drops under a certain relative threshold. With *relative* we mean that we are not using a fixed absolute threshold, but we want to stop when, for example, the Mean Squared Error becomes smaller than a certain fraction of that at the previous iteration. That is:

$$MSE(X^t, X^{t-1}) < \frac{MSE(X^{t-1}, X^{t-2})}{threshold}. \tag{6}$$

This means that we are stopping the algorithm when it is not able to move (or it is moving of a negligible amount) the source point cloud any more; thus, it has converged. We use a relative threshold, instead of an absolute, because it is much more flexible and does not have to be tuned for each set of point clouds. However, rather than checking for Equation (6) just once, we ensure that the condition holds for several consecutive iterations. In this way we avoid stopping too early because of a single iteration during which the alignment was not improved, but that could be followed by other successful iterations.

Another option we evaluated is the use of the so-called *Cost Drop*. During the execution of the inner loop of the multi-iteration version of PPCR, an optimization problem is solved. This optimization problem is a weighted least squares problem, and the cost we want to minimize is the weighted sum of squared distances between corresponding points, as described in the previous section. Before the optimization, the problem we are going to optimize will have a certain cost. The optimizer will, hopefully, reduce this cost to a lower value. The difference between the initial and the final cost is called *Cost Drop*. We use this value to stop the outer loop when the *cost drop* of the inner loop drops under a threshold. We want to avoid absolute thresholds, since they need to be specifically tuned for each application. Instead, we express this threshold with respect to the initial cost of the problem: for example we could stop when the cost drop is less than 1% of the initial cost of the problem. This is what we used for our experiments and proved to be a good threshold for obtaining accurate registrations. This condition is expressed by Equation (7).

$$\frac{cost_{final} - cost_{initial}}{cost_{initial}} < threshold, \tag{7}$$

where $cost_{initial}$ is the initial cost of the optimization problem solved at each iteration of PPCR and $cost_{final}$ is the corresponding cost after the optimization.

Similarly to the MSE, and for the same reasons, this condition should hold for several iterations and not just once.

The third criterion we evaluated is the number of successful iterations of the weighted least squares optimization problems. Solving an optimization problem with Levenberg-Marquard is an iterative process. Each step of this process can be successful, if the step managed to reduce the cost of the problem, or, otherwise, unsuccessful. We wanted to test if this value could be used as termination criterion, for example, stopping after too many unsuccessful steps.

To evaluate the effectiveness of a termination criterion, we used the following idea. Suppose we have the ground truth for the source point cloud, that is, we know the *true* rototranslation between the reference frames of the source and target point clouds. At the end of each iteration, we obtain an estimate of this rototranslation. Therefore, we

can calculate the Mean Squared Error (as in Equation (6)) between our estimate and the ground truth, since they are the same point cloud in different poses. Theoretically, if the algorithm was working properly, this error should decrease among the steps of the outer loop of the algorithm; therefore, the more the iterations, the smaller the difference becomes. Practically, at some point this difference will cease to decrease, or, more precisely, it will start decreasing of a negligible amount. This is the iteration to which we should stop, since it means that the algorithm has converged to a solution. This does not mean that it has converged to the right solution, but, nevertheless, that it is the best solution we can get with the algorithm and the set of parameters we are using.

Ideally, a good termination criterion should behave similarly to the difference w.r.t. the ground truth. It should stop the algorithm more or less at the same iteration at which we would stop if we would be using the difference w.r.t. the ground truth (that, in a real problem, is unknown).

We evaluated the selected termination criteria on two datasets, to find which one works best. Eventually, we evaluated the best one on other datasets, to ensure that the results could be generalized and were not specific to the data we were using for the comparison and that the results we obtained were as good as if we were using a fixed high number of iterations.

## 4. Results

In Figure 2 we plotted the three termination criteria while aligning two point clouds from the Stanford Bunny dataset [27]. The starting transformation between the two clouds is a rotation of 45° around the vertical axis. On the x-axis we have the number of the iteration, while on the y-axis we can find—the number of successful steps of the "inner" optimization problem, the initial and final cost of the "inner" optimization problem, the cost drop (i.e., the difference between the two previous values), the Mean Squared Error w.r.t. the previous iteration, the Mean Squared Error w.r.t. the ground truth and the discrete derivatives of the last three variables. We plotted also the discrete derivatives because they clearly show when a variable is not changing anymore: when the derivative becomes zero, the value of a variable has stabilized.

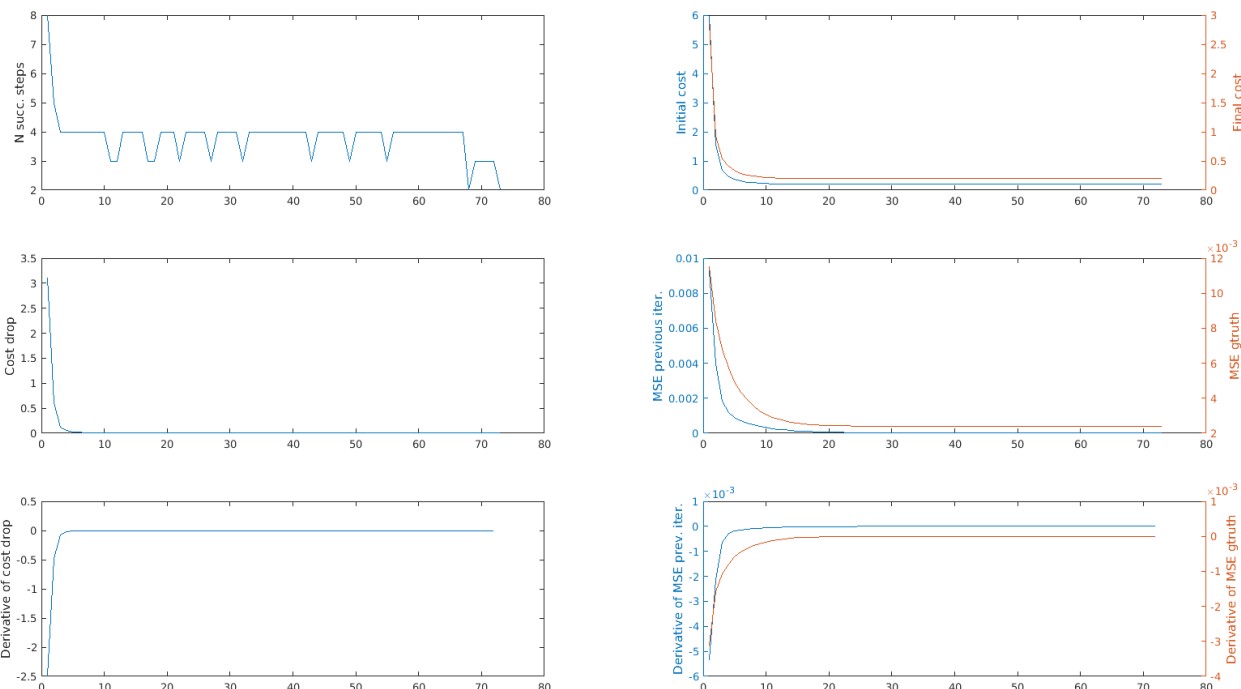

**Figure 2.** Plots of various termination criteria and their derivative for two point clouds from the Stanford Bunny dataset.

We can see that both the cost drop and the MSE w.r.t. the previous iteration have a very similar trend to the MSE w.r.t. the ground truth. Most important, the three values stabilizes more or less at the same iteration. This is particularly obvious if we compare the discrete derivatives: they become almost zero more or less at the same time. Although the MSE w.r.t. to the ground truth keeps decreasing for a few iterations after the other two values stabilizes, its effect on the quality of the result is negligible. This becomes obvious looking at Figure 3, where we have two point clouds, one aligned using a predefined very large number of iterations, the second one using the cost drop as stopping criterion. We can seen that they overlap almost perfectly. The difference between the errors with respect to the ground truth of the two alignments is less than one tenth of the resolution of the point clouds, thus can be considered definitely negligible. Other experiments on the same datasets yielded similar results.

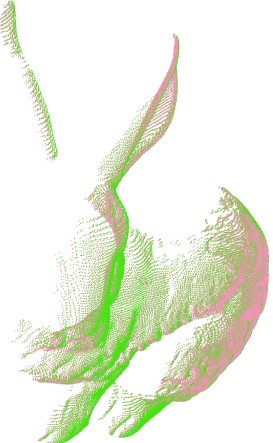

**Figure 3.** The same point cloud aligned with two different termination criteria: a large number of iterations (green point cloud) and our termination criterion based on the cost drop (pink point cloud).

Instead, the number of successful steps oscillates greatly and appears to be not correlated to the MSE w.r.t. the ground truth. For these reasons it was discarded.

In Figures 4a,b we show a pair of point clouds from the Bremen Dataset [28], to which we applied, respectively, a small rotation and a small translation. The results obtained on these data are shown in Figures 5a,b. In these plots, and in the followings, we will not show the derivatives for space reasons. We can see that the cost drop stabilizes more or less when also the MSE w.r.t. the previous iteration stabilizes, that is, when the cloud has already been moved to the right solution (future adjustments are negligible compared to the resolution of the point cloud).

Considering the results, there seems to be no strong reason to choose the MSE w.r.t. the previous iteration over the cost drop as termination criterion. However, it has to be considered that the MSE has to be specifically calculated after each iteration and is relatively computationally intensive, since the whole source point clouds has to be traversed. This is not a computationally expensive operation *per se* but, on the other hand, the relative cost drop is very fast to compute. Indeed, while solving an optimization problem we already calculate the absolute cost drop, since it is used as termination criterion of the inner loop by the optimization algorithm. Thus, calculating the relative cost drop requires only few more operations—it comes practically for free. For this reason we have chosen to use the cost drop as termination criterion—it is very fast to compute and is as good as the Mean Squared Error.

We performed experiments also with clouds that the PPCR algorithm was not able to align properly. The reason is that we wanted to discover whether the termination criteria were able to stop the algorithm early enough, so that computational time is not wasted.

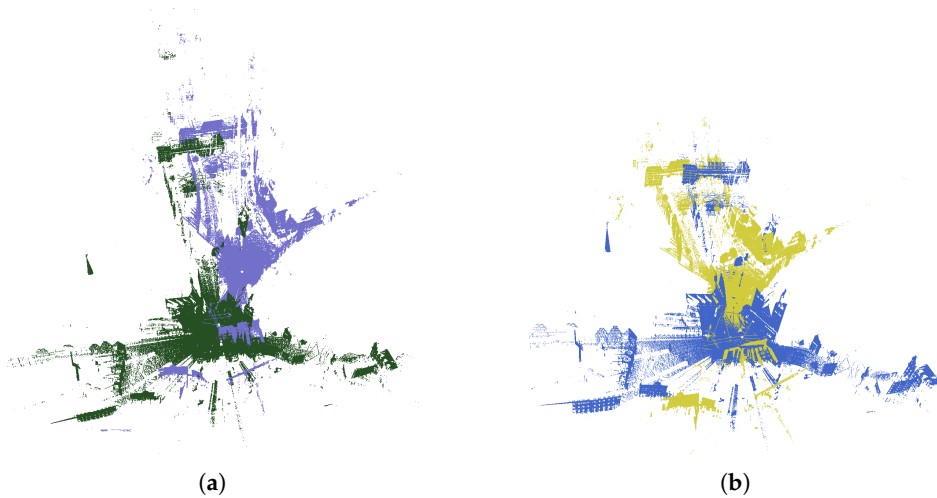

**Figure 4.** Two point clouds from the Bremen Dataset, to which we applied (**a**) a small translation (**b**) a small rotation.

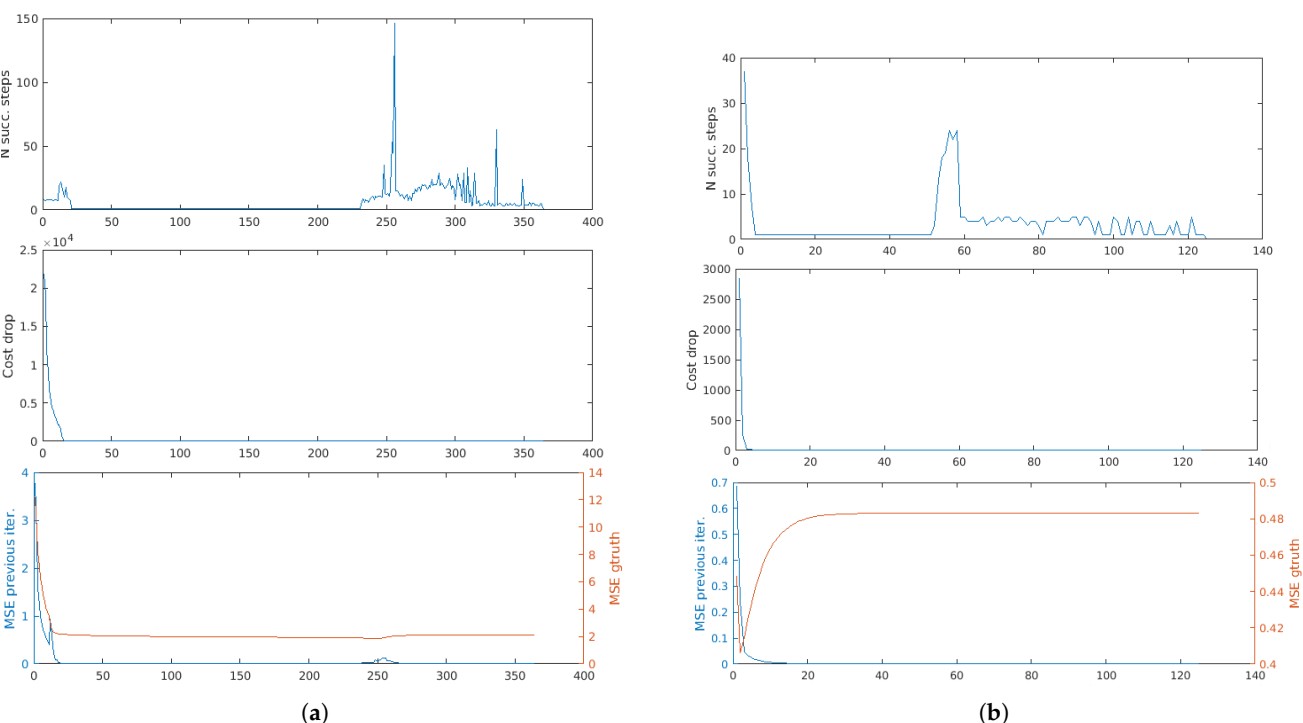

**Figure 5.** Termination criteria for the Bremen Dataset with (**a**) a small rotation (**b**) a small translation applied.

As an example, we show the results on two point clouds from the Stanford Bunny dataset, whose initial misalignment is a rotation of 90° around the vertical axis, (Figure 6a), and a rotation of 180° around the vertical axis (Figure 6b). In these cases, it can be seen that the cost drop stabilizes much earlier than the MSE w.r.t. the ground truth. This behaviour, indeed, is desirable, since it appeared only in unsuccessful alignments, during which stopping earlier is an advantage (going further would be only a waste of computational time).

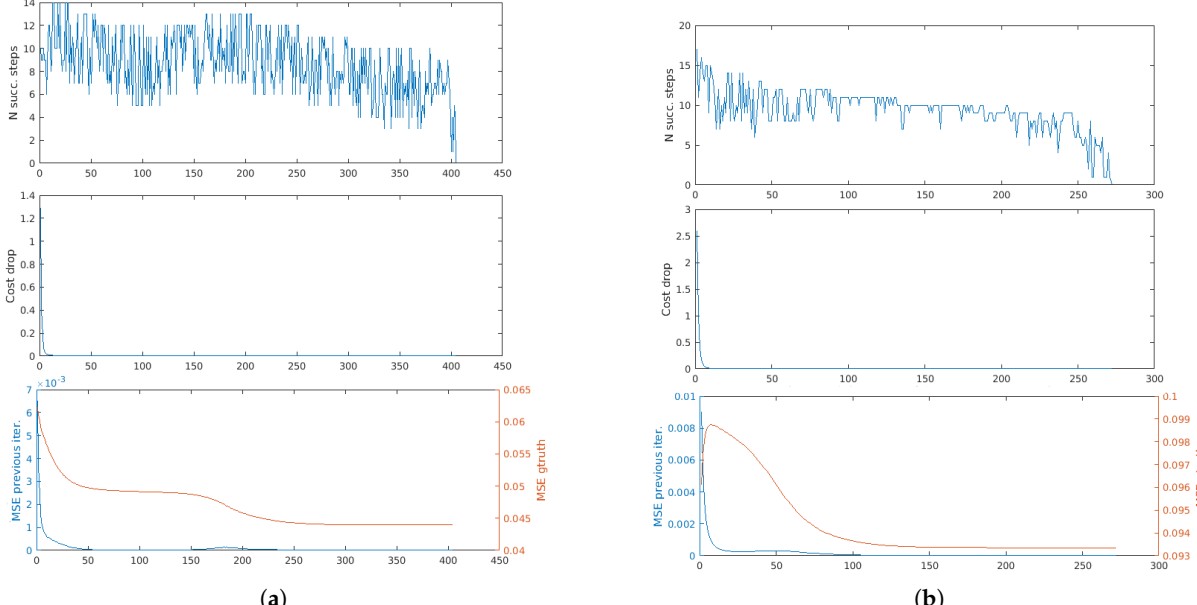

**Figure 6.** Termination criteria for two point clouds from the Stanford Bunny Dataset. Initial misalignment of (**a**) 90° (**b**) 180°.

We tested the chosen termination criterion on the same datasets we presented in our previous work [4] and on pairs of point clouds taken from the Stanford Bunny and Bremen datasets. Our goal is to show that the criterion is effective at stopping the algorithm at the right iteration: too late is a waste of computational time, too early leads to sub-optimal results. For this reason, we did not fine tune other parameters, since the performances of the algorithm were already shown in our previous work.

To show the effectiveness of our termination criteria, we executed the algorithm twice on each dataset. The first time using a predefined very large number of iterations. The second one using the cost drop to stop the algorithm. As a measure of the quality of the results we used the MSE w.r.t. the ground truth. The results are shown in Tables 1 and 2.

**Table 1.** Experimental results with the proposed termination criterion.

| Dataset | MSE | Iterations |
|---|---|---|
| Corridor | 0.31 | 14 |
| Office | 0.42 | 19 |
| Linkoping | 0.53 | 8 |
| Bunny | 0.0025 | 19 |
| Bremen | 0.45 | 8 |

**Table 2.** Experimental results with fixed number of iterations.

| Dataset | MSE | Iterations |
|---|---|---|
| Corridor | 0.66 | 100 |
| Office | 0.48 | 100 |
| Linkoping | 0.50 | 100 |
| Bunny | 0.0024 | 100 |
| Bremen | 0.48 | 100 |

As it can be seen, the results using our criterion are usually comparable, and sometimes better, than when using many more iterations. This means that it succeeds at stopping the algorithm at the right iteration. In some cases, such as for the corridor dataset, the results in Table 1 are much better than those in Table 2. This happens because, sometimes, an excessive number of iterations is not only a waste of time, but could also bring the

algorithm to converge to a wrong solution, even though the right solution was reached. This could happen with every algorithm that uses closest point based associations as a greedy approximation for the (unknown) correspondences.

We performed experiments also on the IRALab Benchmark for Point Clouds Registration algorithms [6]—it is composed of several point clouds, produced with different kinds of sensor and in different environments. Moreover, it includes several registration problems, with different initial misalignments and different overlaps between the clouds to align. For these reasons, we think that is particularly suitable to prove that the chosen criterion is at least as good as using a high number of iterations, but more efficient. Since the benchmark is composed of several datasets, we show statistics for both the single datasets and for the whole benchmark. The result is expressed in terms of median and 0.75 and 0.95 quantiles of the scaled mean squared error, as described in [6]. In Table 3 we compare the median of the results on the various datasets of the benchmark, using PPCR with a high fixed number of iterations (100 iterations) and using the cost drop as termination criterion (stopping when the cost drop is less the 1% of the initial cost for more than 10 iterations). For the cost drop, in the column named *Number of iterations*, we show the mean number of iterations required to solve the registration problems. The same results are shown in Figure 7 as histograms. For most sequences, the differences between the results obtained using the two methods are negligible. Indeed, the medians among all the registration problems of all the sequences (that is, the row named **total** in Table 3) are very close.

However, there are notable exceptions. For the *box_met* and the *urban05* sequences, the cost drop leads to much better results (a lower median means a better alignment). This is the same behaviour we observed for the *Corridor* dataset in the previous set of experiments. On the other hand, on the *p2at_met* and the *plain* sequences the high number of iterations leads to better results. Nevertheless, it has to be considered that, even when the cost drop is not the best termination criterion, its results are still very good. At the same time, the average number of iterations required using the cost drop is 18.55 and, considering the sequences individually, never greater than 31; therefore there is a great reduction in computational time, w.r.t. using 100 iterations.

Table 4 shows that, using the cost drop, we obtain even more consistent results, since its 0.95 quantile is less than that of the 100 iterations. The 0.75 quantiles, instead, are very similar.

The proposed termination criterion requires two parameters—the percentage of drop and the number of iterations during which the condition described by Equation (7) should hold. However, the experiments show that using 1% and 10 iterations as thresholds leads to good results in a very large and varied set of registration problems. Therefore, this values should be adequate for most cases and should not require any further fine-tuning.

In Table 5, we show the results using 1% and 20 iterations as thresholds. The median result is very close to that obtained using 100 iterations, although the mean number of iterations used is less than 30; therefore, there is a great saving in computational time. However, in our opinion, the difference w.r.t. using 10 iterations as threshold is so negligible that it is not worth the extra computational time. Anyway, it is still an option if a very accurate result is desired.

PPCR using the proposed termination criterion, along with instructions on how to use it, is available on GitHub: https://github.com/iralabdisco/probabilistic_point_clouds_registration (accessed on 17 Mar 2021).

**Table 3.** The median of the results of Probabilistic Point Clouds Registration (PPCR) on a comprehensive benchmark [6] with two different termination criterion: 100 iterations and the cost drop. For the cost drop we list also the number of iterations used to reach the solution (under the column *N. of iterations*).

| Sequence | Cost Drop | N. of Iterations | 100 Iterations |
|---|---|---|---|
| box_met | 1.17 | 29.33 | 1.92 |
| hauptgebaude | 0.01 | 31.46 | 0.01 |
| pioneer_slam3 | 0.19 | 24.29 | 0.08 |
| urban05 | 0.36 | 21.81 | 1.13 |
| gazebo_winter | 0.02 | 29.97 | 0.02 |
| planetary_map | 0.59 | 14.82 | 0.42 |
| long_office_household | 0.19 | 23.96 | 0.17 |
| plain | 0.26 | 19.32 | 0.06 |
| pioneer_slam | 0.19 | 27.27 | 0.16 |
| stairs | 0.03 | 24.02 | 0.03 |
| gazebo_summer | 0.06 | 25.11 | 0.04 |
| wood_autumn | 0.02 | 29.34 | 0.02 |
| apartment | 0.07 | 23.04 | 0.06 |
| wood_summer | 0.02 | 30.71 | 0.01 |
| p2at_met | 0.50 | 18.55 | 0.27 |
| total | 0.12 | 18.55 | 0.08 |

**Table 4.** The 0.75 and 0.95 quantiles of the results of PPCR on a comprehensive benchmark [6] with two different termination criterion (100 iterations and the cost drop).

| Sequence | 0.75 Quantile (Cost Drop) | 0.95 Quantile (Cost Drop) | 0.75 Quantile (100 Iterations) | 0.95 Quantile (100 Iterations) |
|---|---|---|---|---|
| box_met | 2.26 | 3.95 | 3.36 | 5.02 |
| hauptgebaude | 0.03 | 0.72 | 0.02 | 0.78 |
| pioneer_slam3 | 0.38 | 0.77 | 0.18 | 0.78 |
| urban05 | 0.50 | 2.12 | 1.77 | 3.33 |
| gazebo_winter | 0.03 | 0.23 | 0.03 | 0.05 |
| planetary_map | 1.16 | 2.18 | 0.83 | 1.81 |
| long_office_household | 0.66 | 2.00 | 0.62 | 2.07 |
| plain | 0.50 | 0.94 | 0.20 | 1.00 |
| pioneer_slam | 0.43 | 3.54 | 0.45 | 4.68 |
| stairs | 0.09 | 0.24 | 0.09 | 0.24 |
| gazebo_summer | 0.20 | 0.65 | 0.13 | 1.02 |
| wood_autumn | 0.03 | 0.27 | 0.03 | 0.04 |
| apartment | 0.29 | 1.30 | 0.27 | 2.02 |
| wood_summer | 0.02 | 0.27 | 0.02 | 0.03 |
| p2at_met | 1.04 | 2.00 | 0.84 | 2.29 |
| total | 0.44 | 1.79 | 0.47 | 2.38 |

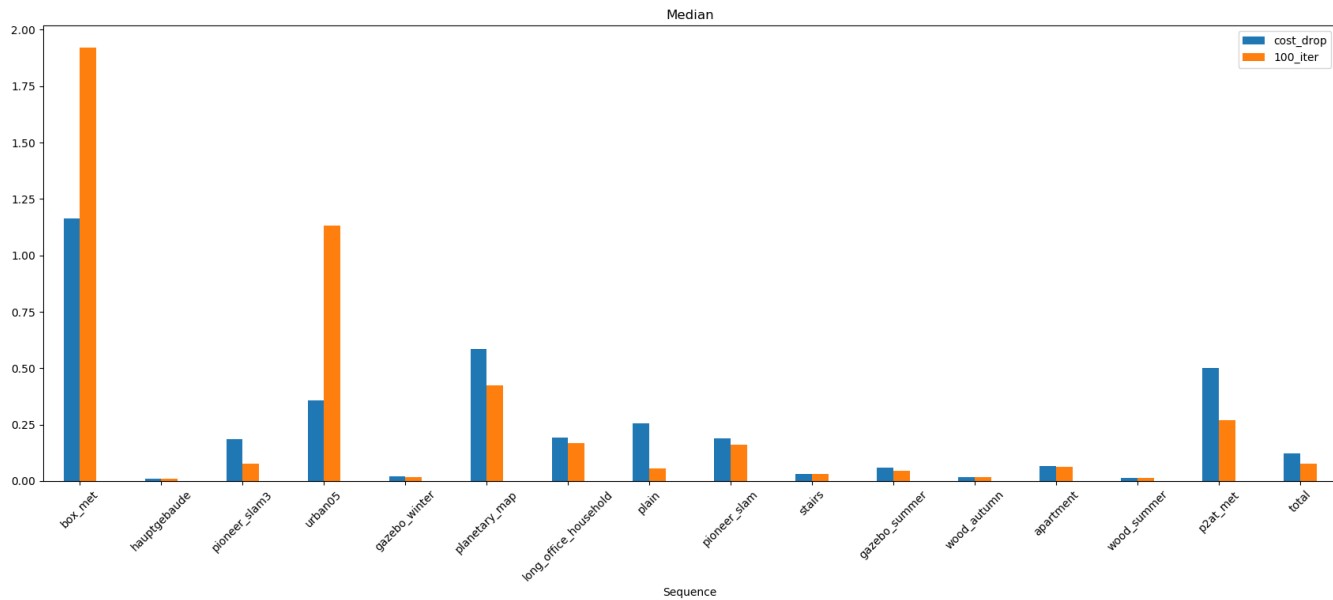

**Figure 7.** Histograms of the median results of PPCR on a comprehensive benchmark [6] with two different termination criterion (100 iterations and the cost drop).

**Table 5.** Results of PPCR on a comprehensive benchmark [6], using the cost drop with 1% and 20 iterations as thresholds.

| Sequence | Median | 0.75 Quantile | 0.95 Quantile | Iterations |
|---|---|---|---|---|
| box_met | 1.28 | 2.49 | 4.27 | 39.40 |
| hauptgebaude | 0.01 | 0.02 | 0.77 | 42.35 |
| pioneer_slam3 | 0.15 | 0.34 | 0.75 | 36.34 |
| urban05 | 0.44 | 0.65 | 4.31 | 39.65 |
| gazebo_winter | 0.02 | 0.03 | 0.05 | 41.09 |
| planetary_map | 0.54 | 1.08 | 2.07 | 24.89 |
| long_office_household | 0.17 | 0.63 | 1.99 | 35.20 |
| plain | 0.17 | 0.46 | 0.93 | 31.78 |
| pioneer_slam | 0.18 | 0.44 | 3.64 | 38.13 |
| stairs | 0.03 | 0.09 | 0.23 | 34.08 |
| gazebo_summer | 0.05 | 0.19 | 0.74 | 35.96 |
| wood_autumn | 0.02 | 0.03 | 0.06 | 40.41 |
| apartment | 0.06 | 0.28 | 1.50 | 33.36 |
| wood_summer | 0.01 | 0.02 | 0.03 | 42.44 |
| p2at_met | 0.47 | 0.97 | 1.96 | 29.75 |
| total | 0.10 | 0.45 | 1.86 | 29.75 |

## 5. Conclusions

We introduced the usage of the relative cost drop as termination criterion for the Probabilistic Point Clouds Registration Algorithm. We tested this criterion on different datasets and on a comprehensive benchmark for point clouds registration algorithms [6], which is composed of several registration problems, with different degrees of overlap and initial misalignment. The experiments prove that the cost drop is effective at stopping the algorithm at the right iteration, that is, when the algorithm has converged to a good solution that cannot be improved substantially anymore. Moreover, it stops the algorithm very early when solving problems that are not going to converge to the right solution even when using more iterations, which is a very desirable behaviour. While it requires two

parameters, we propose values that are effective on a wide range of registration problems and do not need any further fine-tuning.

**Author Contributions:** Conceptualization, methodology, software and writing—original draft preparation S.F.; writing—review, editing, supervision and project administration, D.G.S. All authors have read and agreed to the published version of the manuscript.

**Funding:** This research received no external funding.

**Institutional Review Board Statement:** Not applicable.

**Informed Consent Statement:** Not applicable.

**Data Availability Statement:** The data used in this study is available at https://github.com/iralabdisco/point_clouds_registration_benchmark (accessed on 17 March 2021).

**Conflicts of Interest:** The authors declare no conflict of interest.

**Abbreviations**

The following abbreviations are used in this manuscript:

PPCR     Probabilistic Point Clouds Registration
ICP     Iterative Closest Point
MSE     Mean Squared Error

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
