# Peer review of "A Termination Criterion for Probabilistic Point Clouds Registration"

_signals, 2021_

Round 1

Reviewer 1 Report

Authors build on their work that suggested probabilistic PCR approach with fixed number of iterations. As the parameter is problem-specific, they explore stopping criteria for the Point Cloud Registration(PCR) problem. If such a criterion is found, the iterative PPCR method does not waste cycles if convergence was achieved neither stops before it happens. With the evidence of experiments, the paper suggests that reasonable criteria were suggested, in some cases outperforming the fixed iterations method even though less number of iterations is made.

The paper is well-written and the approach is clearly understood. Novelty merit is low although I believe it will serve as a useful reference for further research.

I have only one minor comment: Based on the mathematical formula I do not agree your policy is based on Euclidean distance as you clam on page 4. Referring to the exponent in probability density of a Gaussian, I claim that Manhattan distance is being used instead. It seems like it still provides satisfactory results, however, given that we are in 3-dimensional space, it might be more reasonable to use exponent 4 instead of 2. Maybe it would improve the results a tiny bit.

Overall, good work.

Author Response

We would like to thank Reviewer1 for his valuable suggestion regarding the exponent of the weighting formula. We will try his suggestion and hopefully get better results! Reviewer1 is certainly right defining the distance we use in the weighting equation as Manhattan instead of Euclidean; however, we think that there has been a misunderstanding. With "data-association policy based on the Euclidean distance" we do not refer to that equation. Instead, we simply mean that we associate to each point in a point cloud, the closest in the other, where "closest" is calculated with a Euclidean distance. The text was probably not clear, so we revised that sentence to reduce chances of any misunderstanding.

Reviewer 2 Report

The paper is focused on the comparison and analysis of different ‘point cloud registration’ termination criteria (3 actually) on datasets to find the best result. The study proposes and used a termination criterion called cost drop. The study uses two roughly aligned point clouds, without using the features of the point clouds, to estimate correspondences. It, however, associates, by weighting, a point in the source point cloud with a set of points in the target point cloud. Hence, the study intended to save computational time with fewer iterations.

The main drawback lies with the abstract, introduction, related work and conclusions, which were not conscientiously written. The author(s) should check the journal’s template and improve the abstract, introduction and conclusion accordingly by giving an overview of the work in general, while highlighting the purpose of the study.

For the related work section, since the study focuses on the termination criteria on datasets, I expected to see literature reviews of termination criteria of algorithms and techniques reviewed. The current review is generic and does not carry a reader in the direction of the purpose of the study. The author(s) should rework the related work section to reflect the termination criteria of algorithms and techniques reviewed, presenting limitations and strengths in them.

The study intends to save computational time with fewer iterations. But the result does not show if this is at the expense of accuracy.

Author Response

We revised the abstract and the introduction sections according to Reviewer2 suggestions. The related work section was expanded including information on other termination criteria and their issues. Moreover, its structure was revised to better highlight the goal of the paper and the issues we want to solve.

We have to disagree regarding the comment "The study intends to save computational time with fewer iterations. But the result does not show if this is at the expense of accuracy.". Indeed, in the experimental section, we perform a very extensive set of experiments on the Point Clouds Registration Benchmark, which show that the median accuracy of the proposed approach is the same as when using a very large number of iterations and, sometimes, even better (as also highlighted by Reviewer1). The set of experiments we use is composed of thousands of registration problems, representing 15 different environments; therefore, we think that we can state that our approach saves computational time without sacrificing the accuracy of the results.

Round 2

Reviewer 2 Report

This is an improved version of the paper. Well done.

Author Response

According to the reviewer's evaluation, we tried to improve the English language, by rewriting sentences that were not so clear and by fixing some minor spelling mistakes.